# Relationships between Psychoeducational Rehabilitation and Health Outcomes—A Systematic Review Focused on Acute Coronary Syndrome

**DOI:** 10.3390/jpm11060440

**Published:** 2021-05-21

**Authors:** Sabina Alexandra Cojocariu, Alexandra Maștaleru, Radu Andy Sascău, Cristian Stătescu, Florin Mitu, Elena Cojocaru, Laura Mihaela Trandafir, Maria-Magdalena Leon-Constantin

**Affiliations:** 1Department of Medical Specialties (I), Faculty of Medicine, “Grigore T. Popa” University of Medicine and Pharmacy, University Street nr 16, 700115 Iasi, Romania; sabina_cojocariu@yahoo.com (S.A.C.); radu.sascau@gmail.com (R.A.S.); cstatescu@gmail.com (C.S.); mitu.florin@yahoo.com (F.M.); leon_mariamagdalena@yahoo.com (M.-M.L.-C.); 2Clinical Rehabilitation Hospital–Cardiovascular Rehabilitation Clinic, Pantelimon Halipa Street nr 14, 700661 Iasi, Romania; 3Institute of Cardiovascular Disease “Prof. Dr. George. I.M. Georgescu”, Address: Carol I Boulevard nr 50, 700503 Iasi, Romania; 4Department of Morphofunctional Sciences I, Grigore T. Popa University of Medicine and Pharmacy, 700115 Iasi, Romania; 5Saint Mary Emergency Medicine Children Hospital, Vasile Lupu Street nr 62, 700309 Iasi, Romania; trandafirlaura@yahoo.com; 6Department of Mother and Child, Grigore T. Popa University of Medicine and Pharmacy, 700115 Iasi, Romania

**Keywords:** myocardial infarction, psychotherapy, education, quality of life, depression, anxiety, physical behavior, illness perception

## Abstract

(1) *Background*: Cardiac rehabilitation is a multidisciplinary program that includes psychoeducational support in addition to physical exercise. Psychoeducational intervention is a component that has had accelerated interest and development in recent decades. The aim was to analyze the current evidence on the effectiveness of psychoeducational interventions for patients with acute coronary syndrome (ACS). (2) *Methods*: We conducted a systematic search of the literature via four databases: PubMed, CENTRAL, PsycINFO, and EMBASE. We included randomized controlled trials that evaluated the effectiveness of a psychoeducational intervention compared to usual care in ACS patients. We assessed the risk of bias using a modified version of the Cochrane tool. We analyzed data regarding the population, intervention, comparator, outcomes, and timing. (3) *Results*: We identified 6248 studies. After a rigorous screening, we included in the analysis 11 articles with a total of 3090 participants. Major adverse cardiovascular events, quality of life, hospitalizations, lipidogram, creatinine, NYHA class, smoking, physical behavior, and emotional state were significantly improved. In addition, illness perception, knowledge, and beliefs were substantially ameliorated (all *p* < 0.001). All this was related to the type and dose of psychological intervention. (4) *Conclusions*: Patients with ACS can receive significant benefits through individualized psychoeducation sessions. The cardiac rehabilitation program should include personalized psychological and educational intervention by type and dose.

## 1. Introduction

### 1.1. Background/Rationale

Acute coronary syndrome (ACS) is the most common disease that requires acute cardiac care, being associated with a high risk of morbidity and mortality and a severe impact on patients and healthcare systems [1]. ACS refers to a spectrum of clinical presentations that include both myocardial infarction (MI) with ST-segment elevation (STEMI) and without ST-segment elevation (NSTEMI) and unstable angina (UA) [2]. The most common pathophysiological mechanism involved is coronary atherosclerotic plaque disruption complicated with thrombosis [3]. Modifiable risk factors for coronary thrombosis are usually hypertension, smoking, diabetes, hyperlipidemia, and obesity, while age, male sex, family history, and ethnicity are the unmodifiable ones [4]. From a morphopathological point of view, the underlying element is cardiomyocyte necrosis for MI, or myocardial ischemia without cell destruction in UA [5]. Cardiac rehabilitation (CR) is a multidisciplinary program whose core components must include psychosocial support and patient education in addition to physical exercise, modification of the cardiovascular risk factors, and dietary counseling [6]. The customization of the program for specific cardiac manifestation started in 2010 when the key steps to deliver CR were established [7]. The latest guidelines of the European Society of Cardiology (ESC) highly recommend that patients with ACS to follow a rehabilitation program [8]. In the modern era, the most notable proven benefits are a 26% reduction in cardiac mortality and an 18% reduction in recurrent hospitalization [9].

Depression and anxiety after an MI develop in 30–40% of patients [10], with both being associated with substantial increases in the risk of adverse cardiovascular outcomes [11,12]. Psychological interventions can reduce the prevalence of emotional disorders, with relaxation training [13], stress management [14], and low-level cognitive behavioral therapy (CBT) techniques [15] being recommended in the CR program. In addition, metacognitive therapy for distressed CR patients might be suitable [16], and problem-solving therapy for depressed patients can also bring additional benefits [17]. Another psychological approach with success in eliciting behavioral change (initiating an exercise regimen and changing dietary habits) [18] and increasing physical activity [19] is motivational interviewing. Associated with psychotherapy, patient education should be integrated into the CR program for it to be complete [20]. Although current data indicate that education-based interventions have no effect on total mortality, total revascularizations, and hospitalizations, the main benefit obtained is the reduction in fatal MI and/or non-fatal cardiovascular events [21].

We consider this systematic review of valuable importance due to its extensive research and, as far as we know, it is the first to evaluate the benefits of psychoeducational rehabilitation focused on acute coronary syndrome. In addition, our paper presents the role of an intervention to prevent emotional disorders such as depression, anxiety, and post-traumatic stress syndrome.

Other systematic reviews in the field provide substantial data on the positive effects of psychological and educational interventions in patients with cardiovascular disease, the most narrow group investigated being one that included patients with coronary artery disease (CAD) [10,21,22,23,24,25,26]. Starting from the fact that the negative psychological effect of acute myocardial infarction is more important compared to that in stable ischemic heart disease [27], we considered it necessary to analyze studies that included only patients with acute coronary syndrome.

### 1.2. Objectives

The aim was to analyze the current evidence on the effectiveness of psychological and educational interventions (as an isolated measure or in a cardiac recovery program) compared to the usual care exclusively for patients with acute coronary syndromes. We aimed to summarize the dose and types of interventions currently administered and their benefits for rehospitalization and quality of life, but also the control of the cardiovascular risk factors, exercise capacity, and adherence to cardiac rehabilitation. We also considered it important to include the understanding and attitude towards the disease, as well as the effects on psychological and medical symptoms.

## 2. Materials and Methods

The protocol was registered on PROSPERO (CRD42021239578). The systematic review was carried out according to the Preferred Reporting Items for Systematic Review and Meta-Analysis (PRISMA) checklist [28], indicated in Appendix A. Additionally, we took into consideration the recommendations from the latest PRISMA statement 2020 [29].

The research question on which we conducted our systematic review was: “Is psychological rehabilitation effective in preventing major adverse cardiovascular events in patients with ACS?”

The population studied in this research consisted of adults with ACS. The intervention was psychoeducational rehabilitation, isolated or in addition to standard cardiac rehabilitation for patients with ACS. The control group consisted of usual care or standard cardiac rehabilitation. The outcomes were: rehospitalization, the quality-of-life evaluation, the control of the cardiovascular risk factors, exercise capacity and adherence to cardiac rehabilitation, the understanding and the attitude towards the disease, and also the effects regarding the psychological and medical symptoms.

### 2.1. Electronic Search Strategy

Relevant publications were indexed by a structured search in the following electronic databases: MEDLINE (Ovid), Cochrane Central Register of Controlled Trials (CENTRAL), PsycINFO (Ovid), and Embase (Ovid). We searched the WHO International Clinical Trials Registry Platform and the US ClinicalTrials.gov Registry for ongoing clinical trials on 15 February 2021. Furthermore, we performed a manual search of the reference lists for the selected articles.

The following search strategy for MEDLINE database was used: (((((((((((“acute coronary”[MeSH Terms] OR (“acute”[All Fields] AND “coronary”[All Fields])) OR “acute coronary”[All Fields]) OR “acute coronary syndrome”[MeSH Terms]) OR ((“acute”[All Fields] AND “coronary”[All Fields]) AND “syndrome”[All Fields])) OR “acute coronary syndrome”[All Fields]) OR (“acute”[All Fields] AND “coronary”[All Fields])) OR “acute coronary”[All Fields]) OR “myocardial infarction”[MeSH Terms]) OR (“myocardial”[All Fields] AND “infarction”[All Fields])) OR “myocardial infarction”[All Fields])) AND (((((((((“cardiac rehabilitation”[MeSH Terms] OR (“cardiac”[All Fields] AND “rehabilitation”[All Fields])) OR “cardiac rehabilitation”[All Fields])) OR ((“interpersonal counseling”[MeSH Terms] OR (“interpersonal”[All Fields] AND “counseling”[All Fields])) OR “interpersonal counseling”[All Fields])) OR ((“mental fitness”[MeSH Terms] OR (“mental”[All Fields] AND “fitness”[All Fields])) OR “mental fitness”[All Fields])) OR ((“positive psychology”[MeSH Terms] OR (“positive”[All Fields] AND “psychology” [All Fields])) OR “positive psychology”[All Fields])) OR ((“motivational interviewing”[MeSH Terms] OR (“motivational”[All Fields] AND “interviewing”[All Fields])) OR “motivational interviewing”[All Fields])) OR ((“cognitive therapy”[MeSH Terms] OR (“cognitive”[All Fields] AND “therapy”[All Fields])) OR “cognitive therapy”[All Fields]))).

### 2.2. Study Selection

For a clinical trial to be eligible for our systematic review, the studied intervention needed to constitute a psychological or behavioral approach. The interventional group was compared to the usual care group so that the additional benefit of psychological intervention could be evaluated. The following selection criteria were applied:Study type: randomized controlled trials;Language: English;Types of participants: adults of all ages who have been diagnosed with an acute coronary syndrome;Types of interventions: psychotherapy, mental fitness, education during hospitalization for the acute event, interpersonal counseling, short-term psychological intervention, motivational interviewing, and positive psychology. Studies were included if they reported a randomized controlled trial for a psychotherapeutic and educational intervention administered by experienced and trained physicists, psychologists, or nurses for adults of all ages;Outcome: all-cause rehospitalization, quality of life evaluation, the control of the cardiovascular risk factors, the exercise capacity, and the adherence to the cardiac rehabilitation program, but also the understanding and the attitude towards the disease, as well as the effects on psychological and medical symptoms;Follow-up duration: without restrictions; if a study was reported in several publications, all follow-up results were taken into account.

Several exclusion criteria were set: (i) interventions or conditions within a study that were not fully randomized, (ii) studies available only as abstracts, (iii) studies that included patients with stable ischemic heart disease without a history of an acute ischemic event, (iv) studies that evaluated physical exercise or other components of cardiac rehabilitation, (v) studies with an intervention arm with fewer than 30 participants (to avoid unreliable findings), and (vi) dissertations, conference abstracts, and studies with a sample size ≤ 100 patients.

### 2.3. Study Appraisal

Title, abstract, and full-text screening was performed in duplicate by two independent reviewers (S.A.C., A.M.). One author (S.A.C.) automatically extracted the data from the studies using a pre-established data extraction form, which is available on request. Full texts of selected articles were revised and papers from the same single study were collected. Similar to the methods for resolving disagreements during the title and abstract screening, independent reviewers firstly discussed the disagreements they had during the data abstraction. If the debate did not lead to a resolution, a third reviewer (M.M.L.C.) made the final decision on the disagreements.

### 2.4. Data Extraction

Information extracted during data extraction covered: author, year of publication, enrollment place and time, number of study participants, type of population (mean age and gender ratio), population education, diagnosis, time of follow-up, and setting. Regarding the details of the studied interventions, we collected from each article the intervention type and description, the dose (number of sessions, total duration in minutes), and comparator. All outcomes along with the measurement method have been reported. The included results have been presented as *p* values, c-statistic/area under the receiver operating characteristics (AUC/AUROC), percentages, standard deviation, mean or median values, and confidence intervals (CIs).

### 2.5. Bias Assessment and Quality of Evidence

The risk of bias for the included randomized controlled clinical trials was evaluated independently by three reviewers (M.M.L.C., C.S., R.A.S.) using AUB KQ1, a modified version of the Cochrane tool [30]. Thus, we evaluated random sequence generation, allocation concealment, selective reporting, other sources of bias, blinding, and incomplete outcome data. Each of these domains was graded as having a high risk of bias, a low risk of bias, or unclear risk. The inconsistencies of the results between the three reviewers were resolved by a fourth one (F.M.).

## 3. Results

Figure 1 highlights the screening and selection process regarding the articles included in this systematic review. Our search in databases identified 6248 studies. After excluding the duplicate articles and scanning the titles and abstracts, we read 149 full-text articles. Of these, only 11 met the inclusion criteria for our analysis.

The risk of bias scores for individual studies are summarized in Appendix A, and the methodological quality graph is presented in Appendix A, both being reported in the Appendix A. Around 35% of all studies were at a high risk of bias due to other sources of bias, such as groups being unbalanced at baseline and the intention to treat analysis. Fewer than 10% of studies did not provide sufficient methodological detail to allow assessment of possible bias in outcome assessment and other sources. The methods of allocation concealment and attrition bias were unclear for no more than 20% of studies.

The characteristics and demographic data for each study are presented in Table 1.

Out of the 11 studies, six were single-sited: Italy [31,44], Portugal [33], USA [35], Iran [36], and Finland [40] and five took place in multiple centers [32,37,38,39,45]. A total of 3090 patients with AMI were included in the selected randomized controlled trials. Among patients, 2237 (72.39%) were men and 853 (27.61%) were women, with a sex ratio of 2.62. Only one study enrolled mainly women, at 68.78% [32]. The mean age for all the people in the included studies was over 50 years old, with one exception that classified the age as fewer than 60 years and 60–75 years old, without including the mean value [40]; no study specified how many patients were young, how many middle-aged, and how many elderly at the time of diagnosis of ACS. Regarding the intervention, seven studies used conventional techniques [31,32,33,36,38,44,45], one study used the telephone [35], one used an online portal [37], and two studies used a hybrid telephone-based and in-hospital method [39,40] as a means of delivering psychotherapy. Only one study of those that had been set up in a hospital was conducted on an outpatient basis [45], the other six involving continuous hospitalization. The longest follow-up time was five years [43], but the initial design of the study was sized to evaluate one-year outcomes. The shortest follow-up time was 1 month [36].

Davidson et al. [32] showed that the effect of problem-solving therapy on depressive symptoms can be generalized across gender, with a minimal difference between the intervention group and the control group: mean −3.6, 95% CI −7.5 to 0.3, *p* = 0.07 for male patients versus mean −4.0, 95% CI −7.6 to −0.3, *p* = 0.03 for female patients. In addition, the effect proved to be generalizable across ethnic background, without a significant difference between the intervention group and the control group: mean −3.5, 95% CI −7.6 to 0.5, *p* = 0.09 for Hispanic patients versus mean −3.5, 95% CI −7.6 to 0.5, *p* = 0.04 for African American patients.

Norlund et al. [37] showed that internet-based cognitive behavioral therapy had no effect on the Hospital Anxiety and Depression Scale (HADS) total score at 14 weeks’ post-baseline for the main analysis. Furthermore, separate exploratory analyses at follow-up showed that men had a lower HADS total score compared with women (β = −2.04, 95% CI −3.60 to −0.47, *p* = 0.01), and there was a borderline significant reduction in HADS total score per unit increase in age (β = −0.08, 95% CI −0.16 to 0.01, *p* = 0.09). In contrast, both the main analysis and separate exploratory analyses showed no effect of treatment on either HADS-anxiety or HADS-depression subscales.

According to the study published by O’Brien et al. [38], the individualized education session delivered using motivational interviewing techniques was more effective in increasing belief scores in patients with a lower level of education at enrollment compared to those who had a moderate or higher level of education (*p* = 0.014). Regarding knowledge and attitude endpoints of the study, the intervention did not have a significant effect for any of the measured covariates: employment, education, insurance, diabetes, and age.

Oranta et al. [40,41,42] investigated the benefits of interpersonal counseling for depression, distress, and quality of life during an 18-month follow-up. For distress, the intervention tested was more effective in patients under 60 years of age compared to patients over 60 years of age (*p* = 0.033). For the other endpoints, there were no significant differences in effect size between the two age categories.

Roncella et al. [44] had as a primary objective the determination of the combined incidence of new cardiovascular events (including myocardial re-infarction, death, stroke, any revascularization procedure, life-threatening ventricular arrhythmias, and recurrence of typical angina pectoris) during one year of follow-up. The authors reported that short-term psychotherapy had a significantly higher primary endpoint effect in patients with a life-event score >10 calculated by Systematic Coronary Risk Evaluation compared to those who had a life-event score <10 at enrollment (OR = 5.78, 95% CI 1.28 to 26.18).

The previous medical history, familial history, environmental aspects, and support of the patients with ACS may affect the effectiveness of psychoeducational therapies. Regarding this issue, no clinical trial included in the present systematic review reported the effect size of these covariates at follow-up, with only descriptive statistics of baseline participant characteristics being available.

The differences in patient characteristic variables between the intervention and control groups were tested with chi-square tests [31,32,33,34,36,38,40,41,42,43,44,45,46], *t*-tests [31,33,34,36,38,39,43,44,45,46], paired *t*-tests [37], or Fisher’s test [36,39,43,44,45,46]. Repeated measures analysis of variance with a heterogeneous compound symmetry covariance structure was used to test the differences in the changes between the groups, applying Mann–Whitney U tests [38,40,41,42,43,44]. The differences in the changes between the groups and the changes within intervention and control groups were analyzed using binary logistic regression in Chivarino et al. [31], Oranta et al. [40,41,42], and Roncela et al. [43,44].

The details of the intervention studied by each trial are summarized in Table 2. Some psychotherapies were built around the strategy of controlling the perception of an event such as mental fitness [31] and a mindfulness training program [36], and others around solving problems [32], while most psychological interventions consisted of multiple components: education, promoting adaptive coping, and cognitive behavioral strategies [33,37,44]. Among the studies included were interventions based on motivational interviewing techniques in the presence of positive psychology [35] or in its absence [38,39,45]. Only one study looked at interpersonal counseling in patients with MI [40]. Only two trials studied group psychological sessions [31,45], and most followed the effects of individual sessions [32,33,35,36,37,38,39,40]. A single trial administered both individual and group sessions [44]. The total dose of psychotherapy was expressed as total minutes. Among the included studies, the highest dose was 1080 min [36], the lowest dose being 80–160 min [38].

The results of the included studies are summarized in Appendix A and represented in Figure 2, being systematically described as follows:

### 3.1. The Impact on Morbidity

#### 3.1.1. Major Adverse Cardiovascular Events

In the COPES trial, Davidson et al. [32] followed major adverse cardiovascular events (MACE), defined as MI or hospitalization for UA. At 9 months, patients receiving problem-solving therapy had fewer MACE events (three events) compared to those in the control group (10 events), with *p* = 0.047. In the STEP-IN-AMI trial, Roncella, Pristipino et al. [43,44] had as a primary outcome a composite index consisting of reinfarction, death, stroke, revascularization, major adverse cardiac and cerebrovascular events, life-threatening ventricular arrhythmia, and recurrence of typical angina. Roncella et al. [44] followed the patients for one year: patients who received short-term psychotherapy had a significantly reduced incidence of the primary composite endpoint (21 events) compared to the group who received the usual care (35 events), with *p* = 0.0006 and with a 35% reduction in the absolute risk. Pristipino et al. [43] followed the patients for up to 5 years: for both the combined incidence of new cardiovascular events and for the cardiovascular events monitored individually, there was no statistically significant reduction in the intervention group compared to the control group.

#### 3.1.2. New Non-Cardiovascular Events

The STEP-IN-AMI trial showed a marked reduction in newly diagnosed non-cardiac events in the group of patients who received short-term psychotherapy at one year (*p* < 0.0001) according to the results published by Roncella et al. [44], as well as at 5 years (*p* < 0.0001) according to the results published by Pristipino et al. [43]. In addition, the authors highlight the fact that such a significant effect has an important impact in improving the long-term prognosis of a post-ACS patient.

#### 3.1.3. Rehospitalization

Oranta et al. [42] proved that patients who received interpersonal counseling had a significant reduction in the use of any specialized healthcare service compared to usual care (*p* = 0.007). However, the characteristics of the patients proved to be important for the patients’ initiative to seek medical attention. The STEP-IN-AMI trial aimed to assess the total number of hospitalizations, including the number of cardiovascular and non-cardiovascular hospitalizations in patients who received short-term psychotherapy compared to usual care. Roncella et al. [44] demonstrated that patients in the intervention group had a significant one-year reduction in the total number of hospitalizations (*p* = 0.02). Pristipino and collaborators [43] revealed the absence of significant improvement at 5 years for any component of the outcome.

### 3.2. Health-Related Quality of Life

Health-related quality of life (HRQoL) was measured by the MacNew Questionnaire in two of the studies included in the analysis, both with evidence of its benefits through psychological interventions. Sunamura et al. [45] described an improvement in the emotional component (*p* = 0.004) and also in the physical one (*p* = 0.015). Roncella et al. [44] illustrated positive effects on the physical component (*p* = 0.03), a lack of significant enhancement for the other two components (emotional and social), and an absence of overall score improvement. The SF-12 questionnaire was used to quantify the quality of life in two of the included studies. O’Neil et al. [39] and Huffman et al. [35] demonstrated the absence of significant improvement in physical and mental components of HRQoL scores. Chivarino et al. [31] evaluated the quality of life through the World Health Organization Quality of Life–Brief Questionnaire (WHOQOL-Brief). At the 8-month evaluation, significant time*group interactions were noted for total score (*p* < 0.001), physical health (*p* < 0.001), psychological health (*p* < 0.001), social relationships (*p* < 0.001), and environment (*p* = 0.026). Oranta et al. [41] assessed the quality of life using the EuroQol-5D (EQ-5D) questionnaire. Compared to standard care, interpersonal counseling did not improve quality of life after myocardial infarction, but the intervention provided positive effects on quality of life in patients over 60 years old.

### 3.3. Cardiovascular Risk Factors

Chivarino et al. [31] described a substantial association between mental fitness and the following medical variables: systolic blood pressure (*p* = 0.019), heart rate (*p* = 0.023), ventricular ejection fraction (*p* = 0.021), low-density lipoprotein cholesterol (*p* < 0.001), high-density lipoprotein cholesterol (*p* < 0.001), triglycerides (*p* = 0.047), and serum creatinine (*p* = 0.002). The results of the study did not show statistically significant data on diastolic blood pressure, blood glucose, and body mass index. Moreover, the OPTICARE trial [44] strengthened the favorable results of a psychoeducational intervention on total cholesterol (*p* < 0.001) and smoking cessation (*p* < 0.001), without obtaining statistically significant data on SCORE risk score, waist circumference (cm), and systolic blood pressure. Therefore, the two studies obtained contradictory results for systolic blood pressure.

### 3.4. Physical Behavior

Mental fitness had a significant impact on the number of patients who continued physical exercise from enrollment to follow-up (*p* < 0.001) compared to usual care, as demonstrated by Chivarino et al. [31]. Psychoeducational interventions delivered using motivational interviewing techniques have proven their effectiveness in promoting physical activity. Huffman et al. [35] evaluated the impact of positive psychology exercises combined with motivational interviewing on physical activity measured with an accelerometer, with the results obtained being promising: higher moderate-to-vigorous physical activity (MVPA) at 24 weeks (*p* = 0.026) by completing 9–15 more minutes per day and taking 1600–1800 more steps per day in the intervention group compared to the control group. Ter Hoeve et al. [46] described the statistically significant impact of group counseling sessions delivered using a motivational interviewing technique on the volume of physical behavior: higher daily step count (*p* = 0.035, additional 513 steps per 14.5 h of daytime waking hours) and increased time in prolonged MVPA (*p* = 0.002) in the intervention group compared to the control group.

### 3.5. Psychological Variables

#### 3.5.1. Depression and Anxiety

The impact of psychoeducation on emotional states has been evaluated in multiple studies as the main objective, being determined by a series of validated questionnaires. Thus, the Beck Depression Inventory (BDI) was used in three of the analyzed studies, all with substantial evidence regarding the association between intervention and depression relief [32,40,44]. The results of studies that measured depression and anxiety using the HADS were inconsistent. Specifically, Fernandes et al. [33] demonstrated that psychological intervention statistically significantly enhanced both the total score and the two components: anxiety and depression (all *p* < 0.0001). In contrast, Nourlund et al. [37] found that patients in both groups (intervention and control) reported a reduction in the class of depressive symptoms, with no difference between the two groups during follow-up. In addition to the two depression assessment scales, O’Brien et al. [38] noted the absence of depressive symptom improvement when measured with the Cardiac Depression Scale (CDS), but reported a statistically significant effect when evaluated with Patient Health Questionnaire 9 (PHQ 9), with *p* = 0.025. Anxiety was given as an independent outcome of depression in two of the studies, with the results being contradictory. More specifically, for anxiety measured using the Cardiac Anxiety Questionnaire [37], psychotherapy did not provide a significant outcome, while for anxiety assessed using the Anxiety Score [45], the intervention was associated with favorable results (*p* = 0.036).

#### 3.5.2. Distress

Nasiri et al. [36] demonstrated that a mindfulness-based training program was associated with decreased stress levels measured with the Perceived Stress Scale-14 (*p* < 0.001) two months after the intervention. In contrast, studies that assessed stress through Symptom Checklist-25 did not obtain statistically significant results when evaluated 12 months after surgery [44] or at the 18-month evaluation [40].

#### 3.5.3. Positive Affect

Huffmann et al. [35] evaluated the outcome regarding positive psychology exercises combined with motivational interviewing on positive affect, a variable measured using the Positive and Negative Affect Schedule (PANAS). In essence, the authors demonstrated a statistically significant association between intervention and improvement in positive affect (*p* < 0.001). The authors suggested that increasing positive affect in post-ACS patients may have important implications, proving the association between positive affect and lower risk of overall mortality in healthy persons and chronic illnesses such as HIV and diabetes [47,48].

#### 3.5.4. Coping Strategies, Self-Esteem, and Health Locus of Control

Chivarino et al. [31] aimed to evaluate coping strategies, self-esteem, and health locus of control, which they measured using Brief Coping Orientations to Experienced Problems (Brief-COPE), General Self-Efficacy Scale (GSES), and Multidimensional Health Locus of Control Scale-form C (MHLC-C), respectively. The results were statistically significant regarding the association between mental fitness and coping strategies, both for the total score (*p* = 0.027) and for two of the three components: emotion-focused (*p* = 0.001) and problem-focused subscales (*p* = 0.002). A substantial outcome was also observed in the relationship between intervention and health locus of control (*p* = 0.002). A significant effect regarding self-esteem was not observed (all *p* > 0.652).

### 3.6. Illness Variables

#### 3.6.1. Cardiac Symptomatology

The NYHA class improved significantly in patients who received short-term psychotherapy compared to usual care at one-year follow-up (*p* = 0.01) as demonstrated by Roncella et al. [44], as well as a five-year follow-up (*p* = 0.01) as Pristipino et al. [43] described. Although the participants in the intervention group experienced a better NYHA class, the echocardiographic parameters (ejection fraction and wall motion score index) were similar to the control group during follow-up. In addition, the authors suggested that psychotherapy influences the perception of the severity of symptoms rather than the actual degree of dyspnea.

#### 3.6.2. Illness Perception

Nasiri et al. [36] showed that the mean score of illness perception assessed using the Brief Illness Perception Questionnaire (BIPQ) was substantially higher in patients in the intervention group (mindfulness-based training program) than in the control group (*p* < 0.001). Fernandes et al. [33] evaluated illness cognition through Portuguese versions of the BIPQ. Regarding illness representations, the term defines the patient’s perception of illness consequences, timeline, experience of symptoms, emotions, concern, personal control, and comprehensibility. The study proved significant time/group interaction effects for all dimensions of the illness representations (all *p* < 0.001). More specifically, patients in the intervention group (brief psychological intervention in phase I of cardiac rehabilitation) perceived fewer negative events in relation to their disease and more positive events for their maintenance throughout the follow-up period. In contrast, patients in the control group showed an increase in perception after discharge of the negative consequences concerning their disease.

#### 3.6.3. Knowledge, Attitude, and Beliefs about Illness

Knowledge was evaluated in two of the trials included in our analysis, both with significant results regarding the beneficial outcome of the studied intervention. Thus, O’Brien et al. [38] demonstrated a substantial effect of individualized educational intervention on knowledge (*p* < 0.001), attitude (*p* = 0.003), and belief (*p* < 0.001) about ACS. In addition, Fernandes et al. [34] proved a notable impact of psychoeducational intervention in improving knowledge about the disease and maintaining it throughout the follow-up (*p* = 0.000).

## 4. Discussion

### 4.1. Summary of Evidence

To our knowledge, this systematic review is the first that aimed to summarize the evidence regarding the impact of psychoeducational rehabilitation in patients with ACS. More specifically, the provided data are clearly promising in terms of the utility of these interventions to improve hard endpoints as well as the quality of life, including alleviation of symptoms of depression and anxiety. Taking into account the heterogeneity issue of the included studies, we emphasize the need for large RCTs with structured integrated multi-modality psychological interventions with a detailed methodology of implementation. By presenting in detail the interventions used in the included randomized controlled trials (type of psychotherapy, number of sessions, and total dose performed) and thus by exposing their heterogeneity, we propose a personalized medicine approach in the psychoeducational rehabilitation of ACS. The benefits of psychoeducational interventions on different aspects of cardiac rehabilitation programs are illustrated in Figure 3.

Most studies in the field enrolled patients without a determination of various mental health comorbidities with ACS before the intervention. The literature presents a minority of trials that divided the intervention group into two subgroups: with and without comorbidity. We did not find any RCT that studied the benefits of a psychoeducational intervention applied only to patients without a mental disorder diagnosed with ACS. This is an important area for future research, taking into account a substantial increase in the prevalence of mental health disorders among patients with acute MI, according to Sreenivasan et al. [49]. Particularly for depression, a multifaceted and bidirectional relationship with cardiovascular disease is described, especially with ACS [50]. Thus, depression by itself may be the cause of MI, but it is not known whether psychoeducation in this category of patients has similar benefits to the same intervention in patients without depressive symptoms. Meta-analyses in this field have demonstrated the benefit of psychological intervention on mortality and morbidity in CAD [11,23,24,25,26,27]. The latest update of the most rigorous reviews (by Cochrane Collaboration) [25] showed the benefit in the current era of optimal psychoeducational intervention. Regarding prognostic outcomes, the positive clinical outcome that resulted from the analysis was for cardiac mortality (RR 0.79, 95% CI 0.63 to 0.98). In contrast, no obvious effect was demonstrated in terms of risk reduction for total mortality (RR 0.90, 95% CI 0.77 to 1.05), rates of revascularization (RR 0.94, 95% CI 0.81 to 1.11), and rates of non-fatal MI (RR 0.82, 95% CI 0.64 to 1.05). The meta-analysis has revealed a reduction in depressive symptoms (SMD −0.27, 95% CI −0.39 to −0.15), anxiety (SMD −0.24, 95% CI −0.38 to −0.09), and stress (SMD −0.56, 95% CI −0.88 to −0.24) in the intervention group compared to the comparator group. In addition, by direct comparison of the studies, the authors demonstrated positive effects on health-related quality of life, type A behavior, and vital exhaustion. Moreover, the systematic review of Reid et al. [11] completes the data from the literature and shows the benefits of psychological intervention on blood pressure for patients. Furthermore, the authors describe a positive effect on knowledge and satisfaction for both patients and their partners. As demonstrated in the Cochrane review [22], education-based intervention in CAD reduced fatal and/or non-fatal cardiovascular events (other than MI) compared to control groups receiving no education (RR 0.36, 95% CI 0.23 to 0.56). Regarding the health-related quality of life, the heterogeneity of measures applied in the studies included in this meta-analysis made it impossible to find consistent evidence. However, there is limited information regarding the improvement of some domain scores. There was no difference in the outcomes for total mortality, fatal and/or non-fatal MI, total revascularizations, and hospitalizations.

### 4.2. Limitations

This paper has some limitations. First, the group of patients analyzed had a defective distribution between the two genders (male/female sex ratio = 2.62), which limits the generalization to the general population. This is due to the fact that sex is a risk factor for CAD, including ACS [51] which led to a preponderance of male patients in our study [52]. Secondly, we searched in the databases only for articles published in English. Thirdly, the comparison with placebo does not apply to psychological and educational interventions, and in all trials the control group was the usual care one. Thus, the nonspecific effects of psychotherapy and education were not accounted for. Fourth, we did not investigate intervention for other emotional disorders such as bipolar disorder. As in any systematic review, there may be a publication bias and the overall picture may be based on positive results, it being known that unsuccessful studies do not end up being published. Finally, there are some important gaps in the literature. It is important to emphasize that the findings are limited by the paucity of randomized controlled trials that have studied psychoeducational intervention exclusively in patients with ACS. Psychoeducational interventions were varied in terms of the type of intervention, the number of sessions and the total duration, the enrolled population, and the setting (phone or in person or both). Therefore, there is no possibility of a meta-analysis, mainly due to the heterogeneity of outcomes and their measurement tools.

## 5. Conclusions and Future Perspectives

Psychoeducational rehabilitation appears valuable in ACS, being associated with improvement in new non-cardiovascular events, quality of life, most cardiovascular risk factors, physical behavior, and mental health outcomes such as depression, anxiety, and distress, along with illness perception and cognitions. In contrast, most interventions proved a lack of enhancing diastolic blood pressure, blood glucose level, body mass index, abdominal circumference, and self-esteem. MACEs and rehospitalizations at 1 year after psychotherapies were significantly reduced, but this improvement was not maintained at 5-year follow-up.

In the era of personalized medicine, patients with ACS should benefit from specific psychoeducational strategies and the choice of the type of intervention should be chosen in accordance with the evidence-based guidelines. Unfortunately, there are currently a limited number of clinical trials that have studied the effect of psychoeducation focused on MI. Taking into account the heterogeneity issue of these studies available in the literature at the moment, we highlight the need for large RCTs with structured integrated multi-modality psychological interventions with a detailed methodology of implementation. Moreover, there is a critical need to establish a number of sessions and a total dose standardized by experts in the field, but this requires further studies. Given the possible health consequences and significant costs of untreated emotional disorders (especially depression) in patients with heart disease, there is a necessity for RCTs to evaluate the impact of psychotherapy on cardiac morbidity and mortality.

## Figures and Tables

**Figure 1 jpm-11-00440-f001:**
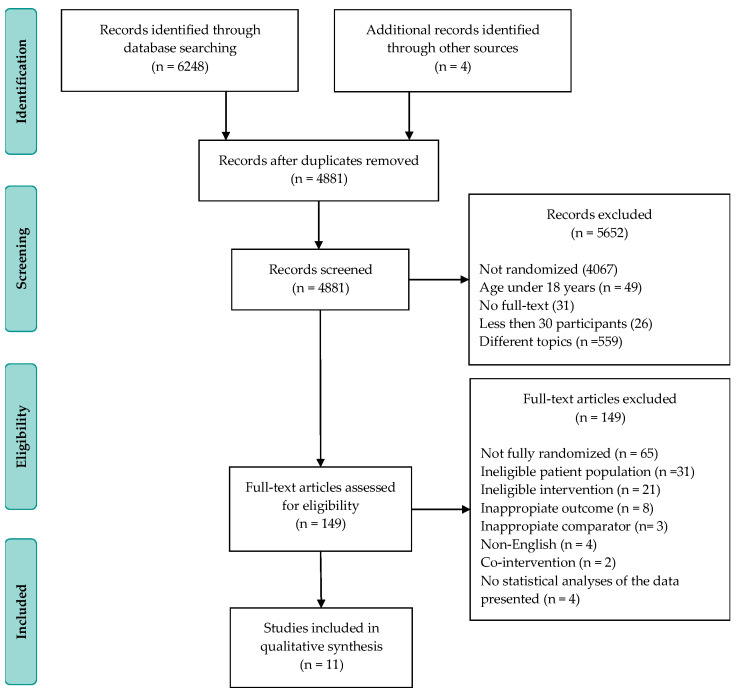
Flow diagram showing selection process.

**Figure 2 jpm-11-00440-f002:**
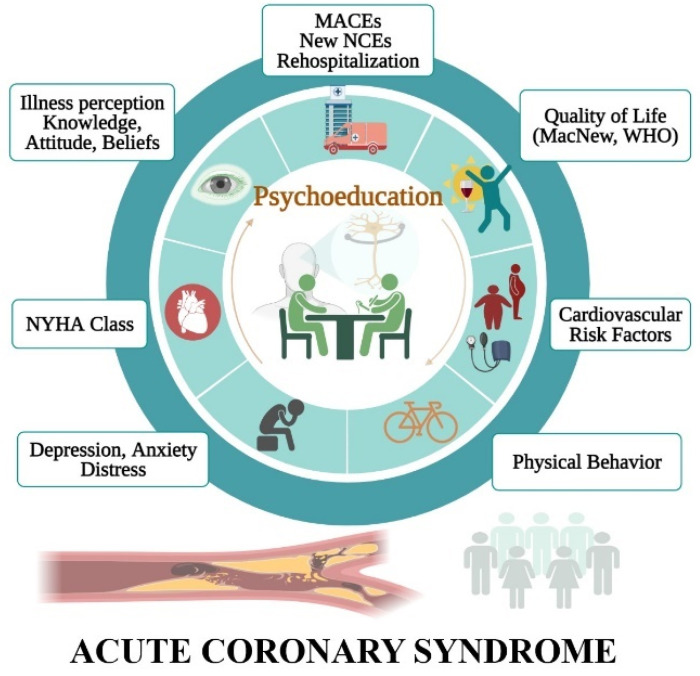
Benefits of psychoeducation rehabilitation focused on acute coronary syndrome.

**Figure 3 jpm-11-00440-f003:**
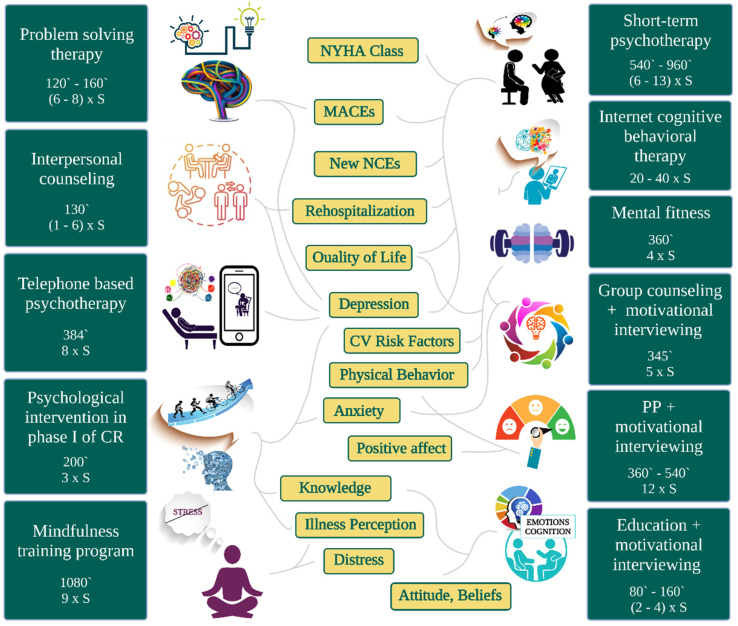
The benefits of psychoeducational interventions specific to the type of cardiac rehabilitation programs and their total dose expressed in number of sessions (S) and minutes (`). CR = cardiac rehabilitation, PP = positive psychology, NCEs = non-cardiovascular events.

**Table 1 jpm-11-00440-t001:** General characteristics of studies included in present systematic review.

Authors and Year of Publication	Enrollment Place (No. of Centers) and Time	No. of Patients F/M (No.)	Mean Age Years (SD)	Education (Mean, No.)	Timing and Setting
Chiavarino et al., 2016 [31]	Italy (1) 2-year period	118 17/101	56.5 (8.70)	10.3 ± 4.0 years	8 months Hospital
Davidson et al., 2010 [32]	USA (5) Between 1 January 2005 and 29 February 2008	157 108/49	61.2 (10.6)	13.1 ± 3.8 years	15 months Hospital
Fernandes et al., 2017, 2018 [33,34]	Portugal (1) 6-month period	121 37/84	61.77 (12.11) versus 66.11 (12.61)	<4 years: 25 4 years: 50 4–12 years: 31 >12 years: 15	2 months Hospital
Huffman et al., 2019 [35]	USA (1) Between May 2017 and April 2018	47 11/36	60.80 (10.7)	Not specified	6 months Telephone
Nasiri et al., 2020 [36]	Iran (1) Between September 2018 and July 2019	64 26/38	52.7 (10.94)	Elementary: 12 Cycle degree: 18 Diploma: 14 Associate degree: 2 Bachelor’s degree: 10	1 month Hospital
Norlund et al., 2018 [37]	Sweden (25) Between September 2013 and December 2016	239 80/159	58.4 (9.0) versus 60.8 (7.8)	Elementary: 48 High school: 91 University: 100	3.5 months Internet-based portal
O’Brien et al., 2014 [38]	Dublin (5) Between October 2007 and October 2009	1136 316/820	62.65 (12.3)	Little formal/primary: 404 Second level: 509 Third level: 222	12 months Hospital
O’Neil et al., 2015 [39]	Australia (6) Between December 2009 and February 2011	121 30/91	61.0 (10.2) versus 58.9 (10.7)	High School: 67 Diploma/trade: 23 Bachelor’s/Master’s: 19	12 months Hospital Telephone
Oranta et al., 2010–1012 [40,41,42]	Finland (1) Between September 2004 and January 2007	103 30/73	< 60 years: 45 60–75 years: 58	Professional Education: 41 Grade II Education: 39 College-level Education: 18 University Education: 5 Profession Worker: 62 Official: 25 Businessman: 16	18 months Hospital Telephone
Pristipino et al., 2019 [43]	Italy (1) Between June 2005 and January 2011	45 10/35	55 (9) versus 55 (8)	Not specified	5 years Hospital
Roncella et al., 2013 [44]					12 months Hospital
Sunamura et al., 2017 [45]	Netherlands (10) Between November 2011 and August 2014	615 124/491	57.5 (9.2) versus 57.4 (9.3)	Low = 19 Intermediate = 319 High = 139	18 months Hospital Outpatient
Ter Hoeve et al., 2018 [46]		324 64/260	58.8 (9) versus 59.1 (9)	Low = 16 Intermediate = 198 High = 78	

**Table 2 jpm-11-00440-t002:** The intervention details for each study (ordered by study ID).

RCT	Type	Description	Delivered by	Dose	Comparator
				Minutes (No. of Sessions)	
Chiavarino et al. [31]	Mental fitness	The sessions were conducted in small groups and lasted 90 min. The intervention was focused on emotions and thoughts. The protocol was based on cognitive theory, being designed for patients with ACS and adapted to the individual power of control of perceptions. The program contained cognitive strategies so that patients were trained to understand and confront the event they were experiencing.	Two specifically trained clinical psychologists	360 (4)	Usual care
Davidson et al. [32]	Problem-solving therapy	The meetings were weekly, in person, or on the phone, each visit lasting 30–45 min. The intervention focused on solving the problem. The protocol was based on increasing the patients’ skills. Participants were taught to assess and expose each psychosocial problem. Pleasant regular activities tailored to each patient were encouraged.	Clinical nurse specialist, psychologist, social worker, and/or psychiatrist	120–160 (6–8)	Usual care
Fernandes et al. [33,34]	Brief psychological intervention in phase I of cardiac rehabilitation	The program was made up of three sessions: education on ACS and cardiac rehabilitation, promotion of psychosocial adjustment in post-ACS rehabilitation (cognitive behavioral strategies for reducing stress and anxiety, education for disease awareness and confidence, promoting adaptive coping, self-monitoring, planning, and family involvement in coping after discharge) and follow-up after hospital discharge.	Session 1: psychologist, cardiologist Sessions 2 and 3: psychologist	200 (3)	Usual care
Huffman et al. [35]	Positive psychology exercises combined with motivational interviewing	The sessions were weekly, delivered by phone, with a duration of 30–45 min each, for a period of 12 weeks. The intervention was composed of two components: a positive psychology component (focused on completing activities based on positive psychology and their application in everyday life) and a motivational interviewing component (used for goal setting to specifically promote physical activity).	Study interventionist	360–540 (12)	Positive psychology exercises alone
Nasiri et al. [36]	Mindfulness training program	The meetings were weekly and lasted 2 h each. The intervention focused on the stress perceived after the acute coronary event and on understanding the disease.	NS	1080 (9)	Usual care
Norlund et al. [37]	Internet-based cognitive behavioral therapy	The intervention included 10 modules with different themes adapted to patients with MI: managing worry, fear, and avoidance, behavioral activation, problem solving, communication skills, applied relaxation training, managing negative thoughts, coping with insomnia, values in life, and relapse prevention. Each module consisted of 2–4 treatment steps. Each treatment stage provided psychoeducation in the form of an electronic text (PDF) along with 1–2 homework assignments. Patients also benefited from additional material and videos that exemplified coping strategies. In addition, patients had access to a discussion board where they could communicate with other patients.	Licensed psychologists	NM (20–40)	Treatment as usual
O’Brien et al. [38]	Individualized education session delivered using motivational interviewing techniques	The meetings were monthly, each visit lasting 40 min. The first session was delivered within 2–4 days of hospital admission at the bedside or in a room off the ward. The intervention consisted of face-to-face education sessions, tailored to the patient’s needs and impact of the disease on the patient’s cognition and emotions. Through motivational training, patients were encouraged to act promptly and appropriately to seek medical attention if required.	NS	80–160 (2–4)	Usual care
O’Neil et al. [39]	Telephone-based psychotherapy	The sessions took place over the phone for 6 months, with an average duration of 48.4 min per session. Intervention sessions were delivered most intensively over the first 3 months. The goal of the program was depression management and cardiovascular risk reduction. The components of the psychological intervention were: motivational interviewing, goal setting, behavioral activation, and cognitive restructuring.	Master’s level qualified psychologists	384 (8)	Usual medical care
Oranta et al. [40,41,42]	Interpersonal counseling	The content of the intervention was modified for MI patients to take from 1 to 6 sessions (mean 4.6, SD 1.24, mode 5), consisting of: starting phase (sessions 1–2): linking the depressive symptoms to the patient’s interpersonal situation and choosing the problem area; encouragement phase (sessions 3–4): working in the problem area, encouragement, processing life changes, finding resources and coping strategies; ending phase (sessions 5–6): encouragement to seek help, encouraging and consolidating the gains, developing ways of identifying and countering depressive symptoms in the future.	Psychiatric nurse trained for one day in the practice of interpersonal counseling	130 (1–6)	Standard care after MI
Pristipino et al. [43] Roncella et al. [44]	Short-term psychotherapy	Individual psychotherapy: 3 to 10 sessions of 1 h with each including personal history elaboration, body language insights, relaxation techniques, and dream analysis. Group psychotherapy: 5 sessions, 2 h each including the same items of individual sessions plus couple analysis, medical/psychological education, and music therapy.	Single psychotherapist	540–960 (6–13)	Usual care
Sunamura et al., 2017 [45] Ter Hoeve et al., 2018 [46]	Group counseling sessions delivered using motivational interviewing technique	The intervention was structured in 3 group counseling sessions, face to face, regarding the physical activity performed. Each session lasted 75 min. In addition, patients participated in 2 more face-to-face group sessions at 3 and 9 months. Each of these sessions consisted of behavioral counseling on heart-healthy lifestyle lasting 1 h per session.	Physiotherapist trained in motivational interviewing	345 (5)	Standard cardiac rehabilitation

RCT: randomized controlled trial, ACS: acute coronary syndrome, MI: myocardial infarction, NM: not measurable: NS: not specified.

## Data Availability

Not applicable.

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
