# Peer review of "Relationships between Psychoeducational Rehabilitation and Health Outcomes—A Systematic Review Focused on Acute Coronary Syndrome"

_jpm, 2021, doi:10.3390/jpm11060440_

Round 1

Reviewer 1 Report

This systematic review aims to evaluate the benefits of psychoeducational rehabilitation in patients with acute coronary syndrome. This manuscript presents the role of an intervention to prevent emotional disorders such as depression, anxiety, and post-traumatic stress syndrome. The analysis was performed on 3090 participants, extracted from 11 studies. Interestingly, the authors found that significant adverse cardiovascular events, quality of life, hospitalizations, lipogram, creatinine, NYHA class, physical behavior, and emotional state were significantly improved. Collectively, this study demonstrates that patients with ACS significantly benefit from psychoeducation sessions and suggest that cardiac rehabilitation programs should include psychological and educational interventions. This is a very interesting. Several issues have been raised. The following points should be addressed:

- The findings from this study are not well illustrated. The authors should consider additional figures to specifically illustrate the benefits of psychological sessions on different aspects of cardiac rehabilitation programs. Substantial efforts towards the illustrations should be considered.

- The authors should discuss or analyze the role of psychoeducational interventions and gender susceptibility. For example, did the authors notice a better response in women or men?

- Do some demographic factors affect the benefits of psychoeducational interventions in ACS patients?  In other words, does the age, socioeconomic or ethnic background affects the benefits of psychoeducational interventions in patients with ACS? This should be further analyzed and discussed in detail.

- The authors should further discuss the contribution of the previous medical history or familial history/experience/environment/support of the patients with ACS as they may affect the effectiveness of psychoeducational therapies in these patients.

- Virtual and in-person sessions are now available. Single-session, multiple sessions, length of the session may also affect the result. How was the analysis normalized to avoid biased conclusions, and how were the control-matched participants chosen for this comparative analysis?

Author Response

Dear reviewer,

Thank you for all your comments.

We did our best to illustrate the benefits of psychological sessions on different aspects of cardiac rehabilitation programs. We used Biorender.com to create Figure 3, being 100% original.

We added some paragraphs regarding the population characteristics of the included studies (gender, age, socioeconomic and ethnic background). We evaluated again all the articles included in our systematic review and none of them report the effect size on these covariates at follow-up, being available only descriptive statistics of baseline participant characteristics. Thank you for this remark, it is very useful for the study.

We have completed the paper with the statistical tests that have been used to avoid biased conclusions.

Hope we have touched all the points you asked us to change.

If there are any other changes you consider we should make, please let us know.

Yours sincerely,

All the authors

Reviewer 2 Report

Reviewer’s comments

In this article on “Relationships between psychoeducational rehabilitation and health outcomes – A systematic review focused on Acute Coronary Syndrome” by Cojocariu et al, the investigators assessed the impact of psychoeducational rehabilitation on various clinical, behavioral, and psychological outcomes among patients hospitalized following an acute coronary syndrome by doing a comprehensive systematic review of all available evidence in the form of randomized control trials till date. I commend the authors for such a detailed review of literature and analysis of individual studies with such a huge heterogeneity to summarize all the evidence on this topic. The novelty in this attempt is that the authors assessed the benefits of psychoeducational rehabilitation among this specific sub-set of high-risk patients with ACS. Even though meta-analysis or any form of pooled analysis was deferred because of study heterogeneity, the article summarizes the individual studies categorized by various outcomes of interest. The article is well written, but I do have a few comments, if addressed promptly, it will improve the manuscript overall.

Major comments

  • In section 4.1, the authors have mentioned they provide conclusive evidence supporting psychoeducational intervention for improving outcomes following ACS. I would be cautious with this argument as the present study only provides a summary of available evidence in this regard without any pooled analysis or meta-analysis because of the extreme heterogeneity of various psychoeducational and/or behavioral interventions employed in various trials. However, the data is clearly promising in terms of the utility of these interventions to improve hard endpoints as well as the quality of life including alleviation of symptoms of depression and anxiety. In this part of discussion and the conclusion sections, authors should emphasize this issue of heterogeneity and need for large RCTs with structured integrated multi-modality psychological interventions with detailed methodology of implementation.
  • In discussion, authors report a key point that they did not find any RCT that studied the benefits of a psychoeducational intervention applied only to patients without depression diagnosed with ACS.

I agree with the authors that the studies separately assessing the utility of psychoeducational or behavioral interventions among patients with ACS with and without known psychopathologies are limited. There is a growing body of evidence that there is a substantial increase in the prevalence of mental health disorders among patients with acute MI (Pls refer to - Mental health disorders among patients with acute myocardial infarction in the United States. American Journal of Preventive Cardiology. 2021;5:100133.). I would recommend authors to stress this aspect of the increasing burden of various mental health comorbidities among ACS and not just depression and mention the need for future studies on the utility of various psychoeducational interventions among ACS patients with detailed sub-group analysis for patients with and without prior mental health disorders.

  • In Methods, rehospitalization is listed as one of the endpoints of interest. Please define rehospitalization as defined in the majority of studies. Is it all-cause rehospitalization or hospitalization for cardiac events?
  • Authors have assessed the risk for bias among included studies. This is reported as supplemental material. Would include a brief note in the main manuscript regarding your overall summary of the assessment of the risk of bias among included studies.
  • In conclusion, line 476-477, the authors report most interventions proved an absence in enhancing MACE and 5-year rehospitalizations… This statement is confusing and contradicting the first part of the conclusion. Please clarify this statement.

Minor comments

  • Line 46 – ACS is the most common disease. Please specify
  • Line 88-89 and line 163 – Would preferably use “stable ischemic heart disease” as a better term for “chronic coronary syndrome”

Author Response

Dear reviewer,

Thank you for all your comments.

In section 4.1, we followed your recommendation and we have changed the conclusive evidence supporting psychoeducational intervention for improving outcomes following ACS. We also changed that in the conclusion section. Thank you for your remark.

As you suggested, we stressed out the fact that there is a growing body of evidence and there is a substantial increase in the prevalence of mental health disorders among patients with acute MI.  We have mentioned this aspect of the increasing burden of various mental health comorbidities among ACS and not just depression and also the need for future studies on the utility of various psychoeducational interventions among ACS patients with detailed sub-group analysis for patients with and without prior mental health disorders.

In the Methods section, we have defined better the rehospitalization for the majority of the included studies.

We included a brief note in the main manuscript regarding the overall summary of the assessment of the risk of bias among the included studies. Thank you for your recommendation.

Regarding the MACEs and 5-year rehospitalizations, we tried to be more specific and clearer regarding these 2 outcomes.

Regarding the two minor comments, we have changed both of them.

Hope we have touched all the points you asked us to change.

If there are any other changes you consider we should make, please let us know.

Yours sincerely,

All the authors

Round 2

Reviewer 1 Report

No further comments. The authors have addressed all my comments and improved the manuscript accordingly.